# Characteristics, Patterns of Care and Predictive Geriatric Factors in Elderly Patients Treated for High-Grade *IDH*-Mutant Gliomas: A French POLA Network Study

**DOI:** 10.3390/cancers14225509

**Published:** 2022-11-09

**Authors:** Coline Montégut, Jean-Sébastien Guillamo, François Ducray, Caroline Dehais, Elisabeth Cohen-Jonathan Moyal, Christine Desenclos, Antoine Petit, Romuald Seizeur, Lien Bekaert, Claude Gaultier, Marie Jeannette Motuo Fotso, Marie Blonski, Jean-Sébastien Frenel, Elodie Vauléon, Olivier Langlois, Georges Noel, Antoine F. Carpentier, Anna Luisa Di Stefano, Charlotte Bronnimann, Dominique Figarella-Branger, Olivier Chinot, Emeline Tabouret

**Affiliations:** 1Neuro-Oncology Department, University Hospital La Timone, 264 Rue Saint Pierre, 13005 Marseille, France; 2Nîmes, Centre Hospitalier Universitaire de Nîmes Caremeau, Service de Neurologie, 30000 Nimes, France; 3Neuro-Oncology Department, Hospices Civils de Lyon, Université Lyon 1, CRCL, UMR Inserm 1052_CNRS 5286, 69000 Lyon, France; 4Service Neurology 2, Mazarin, Assistance Public-Hôpitaux de Paris, Hôpitaux Universitaires La Pitié Salpêtrière-Charles Foix, Sorbonne University, 75013 Paris, France; 5Département de Radiothérapie, Institut Claudius Regaud, Institut Universitaire du Cancer de Toulouse-Oncopole, 31059 Toulouse, France; 6Neurosurgery Department, Hôpital Nord, Centre Hospitalo-Universitaire Amiens, 80000 Amiens, France; 7Neurosurgery Department, Hôpital Jean Minjoz, Centre Hospitalo-Universitaire Besançon, 25000 Besançon, France; 8Neurosurgery Department, Hôpital de la Cavale Blanche, Centre Hospitalo-Universitaire Brest, 29200 Brest, France; 9Neurology Department, Centre Hospitalo-Universitaire de Caen, 14000 Caen, France; 10Neurology Department, Centre Hospitalier Colmar, 68000 Colmar, France; 11Neurosurgery Department, Hôpital Nord, Centre Hospitalo-Universitaire Saint-Étienne, 42270 Saint-Priest en Jarez, France; 12Service de Neurologie, Hopital Central, Centre Hospitalier Régional Universitaire, 54000 Nancy, France; 13Medical Oncology Department, Institut de Cancérologie de l’Ouest, 11 Boulevard Jacques Monod, 44800 Saint-Herblain, France; 14Medical Oncology Department, Centre Eugene Marquis, INSERMU1242, University of Rennes, 35000 Rennes, France; 15Neurosurgery Department, Centre Hospitalo-Universitaire Charles Nicolle, 76000 Rouen, France; 16Radiotherapy Department, Centre Paul Strauss, 76000 Strasbourg, France; 17Neurology Department, Hôpital Saint-Louis, 75010 Paris, France; 18Service de Neurologie, Hôpital Foch, 92150 Suresnes, France; 19Hôpital Saint-André Bordeaux, Centre Hospitalier Universitaire, Service d’Oncologie, 33000 Bordeaux, France; 20Service d’Anatomie Pathologique et de Neuropathologie, Centre Hospitalo-Universitaire Timone, APHM, 13005 Marseille, France; 21GlioME Team, Institute of Neurophysiopathology, Aix-Marseille University 27 Bd Jean Moulin, 13005 Marseille, France

**Keywords:** high grade glioma, *IDH* mutation, elderly, geriatric assessment

## Abstract

**Simple Summary:**

Gliomas remain the most common primary brain tumor in adults. Although they are classified based on the IDH mutation status very little is known considering this alteration in the elderly glioma population. Because IDH-mutated gliomas are associated with better prognosis in the young population, it is essential to characterize its role in elderly and more frail patients to help physicians’ therapeutic decisions. In this study, we demonstrated that elderly IDH-mutated gliomas had very similar characteristics to those found in the younger population but were significantly different from elderly IDH wild-type gliomas. However, patient management in this population appeared to be suboptimal, with less frequent gross total resection and irradiation. We showed that an optimal therapeutic combination of radio-chemotherapy could be safe and feasible for these elderly patients to aid in their management. Finally, we identified specific geriatric prognostic factors such as mobility, neuropsychological disorders, body mass index, and autonomy that can help physicians make future therapeutic decisions for this specific elderly population with a better prognosis.

**Abstract:**

Background: Describe the characteristics, patterns of care, and predictive geriatric factors of elderly patients with IDHm high-grade glioma (HGG) included in the French POLA network. Material and Methods: The characteristics of elderly (≥70 years) patients IDHm HGG were compared to those of younger patients IDHm HGG (<70 years) and of elderly patients IDHwt HGG. Geriatric features were collected. Results: Out of 1433 HGG patients included, 119 (8.3%) were ≥70 years. Among them, 39 presented with *IDHm* HGG. The main characteristics of elderly IDHm HGG were different from those of elderly IDHwt HGG but similar to those of younger *IDHm* HGG. In contrast, their therapeutic management was different from those of younger *IDHm* HGG with less frequent gross total resection and radiotherapy. The median progression-free survival (PFS) and overall survival (OS) were longer for elderly patients *IDHm* HGG (29.3 months and 62.1 months) than elderly patients *IDHwt* HGG (8.3 months and 13.3 months) but shorter than those of younger patients *IDHm* HGG (69.1 months and not reached). Geriatric factors associated with PFS and OS were mobility, neuropsychological disorders, body mass index, and autonomy. Geriatric factors associated with PFS and OS were mobility, neuropsychological disorders, and body mass index, and autonomy. Conclusion: the outcome of *IDHm* HGG in elderly patients is better than that of *IDHwt* HGG. Geriatric assessment may be particularly important to optimally manage these patients.

## 1. Introduction

Gliomas are the most prevalent and aggressive primary brain tumors in adults. They are classified based on the revised World Health Organization (WHO) classification of 2021, they are divided into three histomolecular subgroups based on the presence of isocitrate dehydrogenase 1/2 (IDH1/2) mutations and 1p19q codeletion. Adult diffuse gliomas are divided into IDH-mutated (IDHm) 1p19q codeleted oligodendroglioma, IDH-mutated astrocytoma, and IDH wild-type (IDHwt) glioblastoma [1]. Younger patients are more likely to have IDH mutations, which also provide a better prognosis [2,3]. Currently, recommendations for grade 2 and 3 IDHm gliomas are based on the combination of radiotherapy and chemotherapy by temozolomide [4] or procarbazine, CCNU, and vincristine for patient treatment (PCV) [5,6]. In particular, for patients with anaplastic oligodendroglioma, an ongoing phase III clinical trial questions the use of radiotherapy as first-line treatment in favor of a PCV-only regimen due to their better prognosis (POLCA trial [NCT02444000], performed in the French ANOCEF group and supported by the POLA network). However, to date, there is no prospective study with elderly IDHm patients, and the available retrospective studies are rare with limitations. As a result, the prognostic effect of IDH mutation is unknown in elderly patients, and it is important to characterize the unique traits and therapeutic options for these patients. Due to an increase in life expectancy, recent data suggests an increase in the number of elderly patients with glioma, particularly attributable to IDHwt glioblastomas [7,8,9]. Along with a low Karnofsky Performance Status and the lack of surgical resection, old age is one of the traditional criteria associated with a bad prognosis for glioma patients [10,11]. The prognostic effect of age could be associated not only with the frailties and comorbidities of elderly patients but also with their suboptimal therapeutic management, highlighting the need to specifically improve their oncological treatments [12,13,14,15,16]. There is still a lack of information regarding the prognostic stratification of the elderly population suffering from a primary brain tumor and a dedicated geriatric scoring system for brain tumor patients is missing. Since 2008, a special program in France has been established for more uniform management of de novo adult high-grade gliomas with an oligodendroglial component. The program inter alia aims to provide a pathological centralized evaluation of the cases and molecular analysis linked to a prospective record of clinical and radiological characteristics of patients. Currently, this network has more than 1500 cases including IDHm and IDHwt gliomas. In this context, the present study aims to describe the features, care practices, and survival-predictive geriatric factors of elderly patients with IDHm HGG included in the French POLA network.

## 2. Patients and Methods

### 2.1. Study Design and Participants

The POLA network is a dedicated program that has been set up for more homogeneous management of de novo adult high-grade glioma with an oligodendroglial component (Prise en charge des OLigodendrogliomes Anaplasiques (POLA network)). The aim of the program inter alia is to provide a pathological centralized review of the cases and centralized molecular analysis. Totally, this program includes a large population of oligodendroglioma, *IDHm* astrocytoma and *IDHwt* glioblastoma sent for pathological review. From September 2008 to November 2017, 1433 patients, who were sent for a central pathological review and included in the French nationwide POLA cohort, were included in the present study.

For all cases, formalin-fixed, paraffin-embedded (FFPE) tumor tissue was available for pathological and immunohistochemical analyses. Medical, radiological, histological data and treatment patterns were also prospectively collected from medical records.

All patients were analyzed and segregated according to their age and their *IDH* mutation status. The cut-off age defining the elderly patients was 70 years or older at the time of diagnosis. Firstly, the characteristics and patterns of care of elderly patients *IDHm* HGG were compared to those of younger patients *IDHm* HGG and to those of elderly patients *IDHwt* HGG. Secondly, we focused on the subgroup of elderly patients *IDHm* HGG and gathered geriatric factors to examine their predictive value on survival and toxicity.

The study was approved by a national ethics committee. Patients prospectively included into the POLA cohort provided their written consent for clinical data collection and genetic analysis according to national and POLA network policies.

### 2.2. Data Collection

For all patients at diagnosis, demographic profile (age and gender), presenting symptoms, radiological characteristics on MRI, histology, type of surgery, post-operative Karnofsky Performance Status (KPS) score, and adjuvant treatments received were prospectively and collected in real-time.

Treatment received in the first line were “wait and scan policy”, radiotherapy (RT) alone, chemotherapy (CT) alone, radiotherapy in combination with chemotherapy (RT-CT), or palliative care. Radiotherapy (RT) included conventional radiation therapy (60 Gy delivered in 30 fractions) or hypo-fractionated radiation therapy (40 Gy in 15 fractions). First-line chemotherapy (CT) treatments included temozolomide (TMZ), procarbazine, lomustine (CCNU), and vincristine schedule (PCV) or bevacizumab alone or in combination with temozolomide.

Geriatric parameters including geriatric scores were retrospectively collected, in POLA-centers, by geriatricians, neuro-oncologist, surgeons, radiotherapists, and clinical research assistants participating in the POLA Network.

Geriatric frailties were detected by the G8 score. This screening tool consists of 8 items: appetite changes, weight loss, mobility skills, neuropsychological disorders, body mass index, number of medications, self-rated health, and patient’s age. The score ranges from 0 to 17 and the cut-off value for an impaired G8 score was ≤14/17 [17,18]. The neuropsychological item gathers both depression and dementia disorders. Depression and or dementia are classified as severe, or moderate by clinical assessment, without specific scales or scores.

Functional status was determined by using the Activities of Daily Living (ADL) scale (impaired < 6) [19] and the Instrumental Activities of Daily Living (IADL) scale (impaired < 4) [20]. Comorbidities were identified using the age-adjusted Charlson’s comorbidity Index (severe score ≥ 5) [21].

Cognitive impairment was assessed using the Mini-Mental State Examination (MMSE) (impaired < 24) [22]. Biological markers recorded were hemoglobin, neutrophils, lymphocytes, platelets, serum albumin, protein c reactive, and serum creatinine at diagnosis.

Concerning feasibility, it was defined as the completion of 6 courses of chemotherapy without early stopping for disease progression, death, or unacceptable toxicity (adverse event related to chemotherapy leading either too early treatment stopping, to a dose delay lasting more than 14 days or more than 2 dose reductions, to an unplanned hospital admission or to death).

Concerning toxicity, treatment dose reductions (at the start or during treatment), chemotherapy delay for toxicity, and treatment discontinuation for toxicity were reported. Adverse events were scored using the Common Toxicity Criteria scale for adverse events version 4.0 (CTCAE v4.0).

*IDH1* and *IDH2* mutations: Automated immunohistochemistry (IHC) was performed on 4-µm-thick FFPE sections with an avidin-biotin-peroxidase complex on Benchmark XT (Ventana Medical System Inc., Tucson AZ, USA) using the Ventana Kit including DAB reagent to search for the expression of *IDH1* R132H (Dianova, H09). When the results of *IDH1* R132H IHC were negative or unreliable, the status of *IDH1* and *IDH2* mutation was assessed by direct sequencing using the Sanger method and primers, as described previously [23].

Tumor DNA was extracted from frozen tissue, if available, or from FFPE samples using the iPrep ChargeSwitch^®^ Forensic Kit. Qualification and quantification of tumor DNA were performed using a NanoVue spectrophotometer and gel electrophoresis, respectively. The genomic profile and assessment of the 1p/19q codeletion status were determined as described previously [24]. When the quantity of DNA was insufficient to perform SNP or CGH, microsatellite analysis was conducted (LOH) of chromosomes 1p and 19q were assessed via PCR techniques described elsewhere [25].

### 2.3. Statistical Analysis

Data are presented as a median, range, mean and standard error of the mean (se). For correlation analysis, the chi-square test (or Fisher’s exact test) was used to compare qualitative variables. Continuous variables were compared using the Mann-Whitney U test. Progression-free survival (PFS) was defined as the time from the date of surgery to recurrence or death from any cause, censored at the date of the last contact. Overall survival (OS) was defined as the time from the date of surgery to death from any cause, censored at the date of the last contact. The Kaplan-Meier method was used to estimate survival distributions. Log-rank tests were used for univariate comparisons. Cox proportional hazards models were used for multivariate analyses and for estimating hazard ratios in survival regression models. The sensitivity and specificity of the brain geriatric score were analyzed using receiver operating characteristic (ROC) curve analysis after dichotomization of patient survival (<48 months vs. ≥48 months). All statistical tests were two-sided, and the threshold for statistical significance was *p* = 0.05. Analyses were conducted using PASW Statistics version 22 (IBM SPSS Inc., Chicago, IL, USA).

## 3. Results

### 3.1. Elderly Patients IDHm HGG

#### 3.1.1. Clinical Characteristics and Patterns of Care

Depending on the age and the presence of *IDH* mutation, patients were divided into four groups: 39 elderly patients *IDHm* HGG; 80 elderly patients *IDHwt* HGG; 933 non-elderly patients *IDHm* HGG and 381 non-elderly patients *IDHwt* HGG (Appendix A).

The clinical, radiological, histological, and treatment characteristics of the elderly patients *IDHm* HGG are summarized in Table 1. The median age at diagnosis was 74 years (range 70.2–87.1 years). At diagnosis, half of the elderly patients *IDHm* HGG presented with epilepsy and a quarter of them with cognitive disorders. Median post-operative Karnofsky Performance Status (KPS) was 80 (range 50–100). Epilepsy was present at diagnosis in 15 patients (44%) and was the only symptom in 9/15 patients (60%). Seizures were polymorphous and independent from onco-geriatric factors and patient survival. Regarding neuroimaging, the majority of patients presented with contrast enhancement (78.8%). Diagnoses consisted of 1p/19q co-deleted anaplastic oligodendroglioma in 72% of cases and *IDHm* grade III or IV astrocytomas in the remaining 28%. Median Ki67 expression was 15% (range: 0–40). Steroids were prescribed during the post-operative period for half of the patients (48.5%). Regarding the treatment approaches, two-thirds of patients had biopsy alone and adjuvant treatments consisted of chemotherapy alone (Table 1 and Appendix A).

#### 3.1.2. Geriatric Characteristics

Out of the 39 *IDHm* HGG elderly patients, geriatric data were available for 34 of them (Table 2). *IDHm* elderly patients HGG had heavy comorbidities and medications as age-adjusted Charlson’s index was ≥5 for 23 patients (72%). Seven patients (28%) presented with neuropsychological disorders, assessed by G8 score. Mobility was preserved for 16 patients (66.7%). Weight loss was experienced by 12 elderly patients (48%) and 7 (26%) met criteria of malnutrition at diagnosis. Elderly patients suffered from autonomy loss as showed by ADL score < 6 for 6 patients (33.3%) and IADL score < 4 for 8 patients (47%). The estimated G8 score was ≤14/17 for 16 patients (64%). Neuropsychological disorders assessed by G8 score, loss of mobility, number of comorbidities, malnutrition, loss of autonomy assessed by ADL score, and impaired G8 score were more frequent in astrocytoma than oligodendroglioma elderly patients (Appendix A).

#### 3.1.3. Feasibility and Safety of Adjuvant Treatment Chemotherapy and Radiotherapy

Regarding the treatment feasibility, 72% of patients treated by TMZ received at least six cycles while 56% of patients treated by PCV completed six cycles (Appendix A). In terms of chemotherapy treatment safety (Appendix A), at treatment initiation, seven patients (41%) in the TMZ group and five patients (62%) in the PCV group had a baseline dose reduction due to their age. In the TMZ group only, the dose was increased for four patients for a second time. During treatment, three patients (19%) had a TMZ dose reduction and seven (87%) had a PCV dose reduction. Treatment interruption due to adverse events occurred fortwo patients (12%) treated by TMZ and six patients (75%) treated by PCV (Appendix A). In total, grade 3 or 4 adverse events occurred for three patients treated by TMZ and two patients treated by PCV patients (Appendix A).

Regarding radiotherapy treatment safety, only one treatment interruption was reported due to toxicity (grade 2 asthenia) and only one patient had an early serious adverse event (post-radiation encephalopathy two months after the end of RT).

Regarding predictive factors of treatment toxicity, a low post-operative KPS and a poor self-reported state of health were correlated with a higher grade 3–4 toxicity probability. The loss of mobility was predictive of treatment interruption due to toxicity (*p* = 0.047) (Appendix A).

### 3.2. Characteristics and Patterns of Care Group Comparisons and Outcomes

#### 3.2.1. Comparison between Elderly Patients IDHm HGG (*n* = 39) and IDHwt HGG (*n* = 80)

Clinical, radiological, and histological presentations of elderly patients *IDHm HGG* significantly differed from those of elderly patients *IDHwt HGG* (Table 1). Elderly patients *IDHm* HGG presented with less frequent mnesic disorders (11.4% vs. 37%, *p* = 0.054), radiological necrosis (31.8% vs. 61.2%, *p* = 0.039), contrast enhancement (78.8% vs. 95.8%, *p* = 0.01) and had a lower proliferative index (*p* = 0.005).

#### 3.2.2. Comparison between Elderly (*n* = 39) and Non-Elderly (*n* = 919) Patients IDHm HGG

No difference regarding the clinical, radiological, and histological presentations of elderly and non-elderly patients *IDHm HGG* were observed. In contrast, their managements were significantly different (Table 1). Compared to their younger counterpart, elderly patients *IDHm* HGG less frequently underwent gross total or subtotal resection (24.3% vs. 50.3%, *p* = 0.002) and radiotherapy (48.7% vs. 80%, *p* < 0.001).

#### 3.2.3. Outcomes

The median progression-free survival (PFS) and overall survival (OS) were longer for elderly patients *IDHm* HGG (29.3 months (95% CI 22.1–36.4 months) and 62.1 months (95% CI 13.2–111.1 months) respectively) than elderly patients *IDHwt* HGG (8.3 months (95% CI 6.0–10.6 months) and 13.3 months (95% CI 10.6–16.0 months) respectively, *p* < 0.001) but shorter than those of younger *IDHm* patients HGG (69.1 months (95% CI 59.0–79.1 months) and not reached respectively, *p* < 0.001). Finally, median PFS and OS for elderly patients *IDHm* HGG were longer than those of younger patients *IDHwt* HGG. More precisely, in elderly patients, *IDHm* HGG, PFS, and OS in the oligodendroglioma subgroup were longer than those of astrocytoma (Figure 1). When focusing on patients receiving radiotherapy, we still observed different outcomes between these four subgroups and especially between elderly *IDHm* vs. *IDHwt* HGG patients.

### 3.3. Prognostic Factors in Elderly Patients IDHm HGG

Median PFS was 29.3 months (95% CI 22.1–36.4 months). In univariate analyses (Table 3), median PFS was shorter for elderly patients *IDHm* HGG presenting with a loss of mobility (11.0 months (95% CI 3.75–18.32) vs. 53.7 months (95% CI 28.82–78.68), *p* = 0.003); severe neuropsychological disorders according to G8 score (10.2 months (95% CI 7.70–12.79) vs. 50.0 months (95% CI 19.40–80.69 months), *p* = 0.003); clinical criteria of denutrition (11.0 months (95% CI 9.00–13.07) vs. 50.0 months (95% CI 21.29–78.80), *p* < 0.001); and a loss of autonomy according to ADL score [9.25 months (95% CI 6.51–11.99) vs. 65.19 months (95% CI 31.24–99.13), *p* < 0.001 ). Low post-operative KPS, low hemoglobin serum level, low lymphocyte count, and grade IV glioma were also associated with poor PFS (*p* = 0.012 and *p* = 0.004, respectively). Multivariate analysis adjusted by 1p19q codeletion confirmed that loss of mobility (HR = 4.7 (1.30–17.21), *p* = 0.018), presence of severe neuropsychological disorders (HR = 4.3 (1.56–11.92), *p* = 0.005)), clinical denutrition (HR = 5.26 (1.83–15.04), *p* = 0.002), loss of autonomy according to the ADL score (HR = 14.97 (2.73–81.95), *p* = 0.002 for ADL score), lymphopenia (HR = 4.8 (1.27–18.08), *p* = 0.02) and lower KPS (HR = 3.77 (1.20–11.80), *p* = 0.023) were significantly associated with shorter PFS.

Median OS was 62.1 months (95% CI 13.2–111.1 months). In univariate analyses (Figure 2), median OS was shorter for elderly patients *IDHm* HGG presenting with a loss of mobility (15.50 months (95% CI 0.156–30.85) vs. not reached, *p* < 0.001), severe neuropsychological disorders (22.16 months (95% CI 0–50.0) vs. not reached, *p* = 0.036), clinical criteria of denutrition (22.16 months (95% CI 5.1–39.0) vs. 87 months (95% CI 37.12–136.89), *p* = 0.002) and a loss of autonomy according to ADL score (11.04 months (95% CI 0–27.07) vs. not reached, *p* < 0.001). A low post-operative KPS, grade IV rating, and the absence of 1p19q codeletion were also associated with a poor OS (*p* = 0.002, *p* = 0.001, *p* = 0.043, respectively).

Multivariate analysis adjusted by 1p19q codeletion confirmed that loss of mobility ([HR = 9.93 × (1.82 − 53.98), *p* = 0.008), presence of severe neuropsychological disorders (HR = 3.51 × (1.02 − 12.14), *p* = 0.047), clinical criteria of denutrition (HR = 5.85 × (1.66 − 20.61), *p* = 0.006), loss of autonomy according to the ADL score (HR = 21.56 × (2.11 − 219.86), *p* = 0.01) and lower KPS (HR = 5.3 × (1.5 − 17.95), *p* = 0.007) were correlated with shorter OS. In contrast, the G8 score was not correlated either with PFS or OS.

Finally, focusing on older patients with *IDHm* 1p19q codeleted oligodendroglioma, patients treated by RT-PCV demonstrated longer PFS (*p* = 0.08) and OS (*p* = 0.037) than the patients treated by RT-TMZ (Appendix A).

### 3.4. Towards a Brain Tumor Geriatric Score

Because the classical G8 score was not able to predict elderly patients’ survival, we designed a new geriatric scoring system based on geriatric factors significantly correlated with patient survival in our multivariate analysis. This brain geriatric score (BGS) included the following items: mobility, neuropsychological disorders, and body mass index as evaluated by the G8 score and the ADL score as reported by *Katz* et al. (Figure 3). After dichotomization of patient survival (<48 months vs. ≥48 months), the sensitivity and the specificity of the BGS to predict long-term survival were 100% and 83% in our cohort with an AUC of 0.948 (Figure 3). Using a cut-off of 10, the BGS score was significantly correlated to PFS and OS (*p* < 0.001 both, Figure 3). Median PFS was 50.1 months (95% CI 17.6–82.4) for patients with a BGS ≥ 10 vs. 9.2 months (95% CI 8.1–10.3) for patients with a BGS < 10 (*p* < 0.001). Median OS was not reached for patients with a BGS ≥ 10 vs. 11.0 months (95% CI 6.1—15.9) for patients with a BGS < 10 (*p* < 0.001) (Figure 3).

## 4. Discussion

In the present study, we showed that, albeit infrequent, *IDHm* HGG occurred in a small number of elderly patients. Compared to elderly patients *IDHwt* HGG, elderly patients *IDHm* HGG presented with distinct clinical, radiological, and histological characteristics and a better outcome. In contrast, their baseline characteristics were similar to those of younger patients *IDHm* HGG despite a worse outcome. We subsequently showed that adjuvant treatment feasibility and toxicity were advisable, with limited grade 3–4 toxicity occurrence. Finally, we analyzed the prognostic value of classical geriatric factors and identified four of them as able to segregate two groups of patients with completely different outcomes.

To our knowledge, the extensive characterization of elderly *IDHm* HGG patients was never previously reported with the precision of the present study allowed by the POLA network. Andrews and colleagues [7] evaluated the *IDH* mutation occurrence in a retrospective cohort of 224 patients aged 55 years or older with diffuse gliomas. In this cohort, 42 patients presented with *IDH1* R132H mutations, and 29 patients (13%) presented with minor *IDH* mutations underlining their occurrence after the cut-off defined by the WHO 2016 classification. Based on our results and the favorable outcome of elderly patients *IDHm* HGG, the misestimation of this molecular profile could negatively impact their management and survival.

In the present study, we reported that elderly patients presented with the poorer outcome than their younger counterparts despite the absence of additional pejorative factors, including a worse KPS at diagnosis and a larger tumor size. This difference in survival between younger and older *IDHm* patients cannot be reduced to a simple “age” effect: it seems to be partly explained by a difference in treatment management. In our study, 62% of elderly patients *IDHm* HGG underwent only an initial biopsy (compared to 19% in patients <70 years) and less than half of them were treated by radiotherapy (compared to 80% in patients <70 years). Regarding the surgery approach, several studies suggest that maximal resection in an elderly patient can be safe and is associated with better outcomes [26,27,28]. These studies only included GBM patients but it seems reasonable to extend their conclusion to *IDHm* glioma patients with longer life expectancy. Additionally, we reported very encouraging results of feasibility and safety for adjuvant treatment. In oligodendroglioma patients presenting with longer life expectancy, the feasibility of the PCV regimen encourages proposing this treatment alone when possible, according to the younger patients’ guidelines [5]. In contrast, the use of radiotherapy in this population at higher risk of neurotoxicity could be delayed. For *IDHm* astrocytoma elderly patients, the combination of chemotherapy and irradiation seems feasible and promising. No previous data concerning *IDHm* elderly patients were available, but the feasibility and efficacy of radiotherapy for GBM elderly patients were first demonstrated by Keime et al. [29]. There were no immediate severe adverse events related to RT in their study. Importantly, quality of life and cognitive assessment did not differ between treatment arms. More recently, a phase III clinical trial [30] reported the feasibility and the efficacy of the combination of short-course radiotherapy (40 Gy, 15 fractions) and temozolomide in elderly patients. A total of 562 older patients (>70 years) were treated by RT-TMZ or RT alone. Median OS was higher in the experimental arm and grade ≥ 3 lymphopenia (27.3%), thrombocytopenia (11.1%), and neutropenia (8.3%) were the more frequent adverse events. The association of radiotherapy and chemotherapy appears to be an interesting option for elderly *IDHm* astrocytoma HGG patients, keeping in mind that the risk of delayed neurocognitive disorders after irradiation should be evaluated according to the life expectancy of each patient. In this context, a predictive scoring system to identify patients with long life expectancy would be useful for better patient management. Frailty assessment of oncological patients was previously reported in different large tumor-type cohorts and in specific sub-population [15,16]. In 2011, the G8 score was developed to screen frailty in elderly cancer patients. This tool was designed to detect the vulnerable patients who would be offered comprehensive geriatric assessment, with a sensitivity of 87% and a specificity of 58% [18]. Among its main limitations, we can note a low specificity and the absence of patients with a primary brain tumor in its validation prospective cohort. More specifically in neuro-oncology patients, geriatric factors evaluation was previously reported in retrospective cohorts of GBM patients but never in *IDHm* glioma patients. Ackerl et al. [31], reported a retrospective cohort of 70 patients underlining the wide outcome variability in this population and the importance of geriatric assessment-based therapy management. In their study including 34 elderly patients with newly diagnosed GBM, Giaccherini et al. [32] generated a prognostic score based on the combination of KPS, type of surgery, and Frailty Index. Scheinder et al. [33], demonstrated that modified Frailty Index (≥0.27), comorbidity burden (CCI > 2), and nutritional status (BMI < 30) were significantly associated with poor OS in geriatric patients (≥65 years) with GBM. Lombardi et al. [34], enrolled 113 elderly GBM patients (≥65 years) and reported that comprehensive geriatric assessment was an independent significant predictor of mortality. In a retrospective study, Lorimer et al. [35] demonstrated that geriatric features such as cognition and autonomy impairment were negatively associated with survival in elderly GBM patients. Finally, Deluche et al. [36], reported the first retrospective study demonstrating the feasibility and the prognostic value of the G8 score in 89 elderly patients with GBM. In our study dedicated to elderly *IDHm* HGG patients, the G8 score and the CCI did not correlate with survival at any cut-off value, leading to the generation of a new prognostic factors combination allowing the identification of long-term survivor patients for whom treatment management should be optimized. This score needs now to be validated in an independent larger series. If confirmed, it could be an interesting tool for neuro-oncologist before the initiation of treatment in this population.

The main limitation of our study is the limited number of elderly *IDHm* HGG patients, the retrospective record of specific geriatric features, the use of MMSE for neurocognitive evaluation, and the lack of validation of our findings in an independent cohort. However, to our knowledge, this study represents the largest and the most characterized cohort of elderly *IDHm* HGG in literature. Now, the next steps will be to prospectively validate our results in a dedicated multicentric study, facilitated by the French POLA network.

## 5. Conclusions

To our knowledge, this is the first study characterizing elderly *IDHm* HGG patients. Although rare and heterogeneous, IDHm HGG shows better outcomes in elderly patients than that of *IDHwt* HGG. Geriatric assessment may be particularly important to optimally manage these patients. These results need now to be prospectively confirmed.

## Figures and Tables

**Figure 1 cancers-14-05509-f001:**
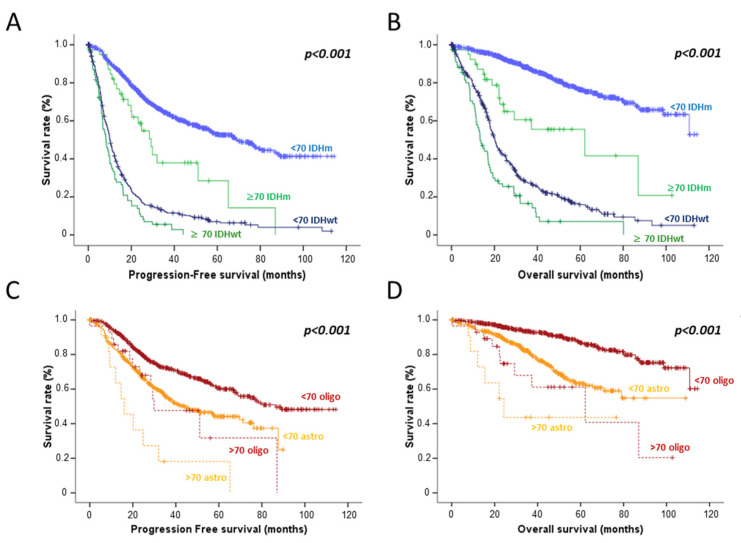
Progression free-survival (**A**) and overall survival (**B**) according to the age (< or ≥70 years) and *IDH* mutation status. Progression free-survival (**C**) and overall survival (**D**) according to the age and the histological subtype of *IDH* mutated HGG patients.

**Figure 2 cancers-14-05509-f002:**
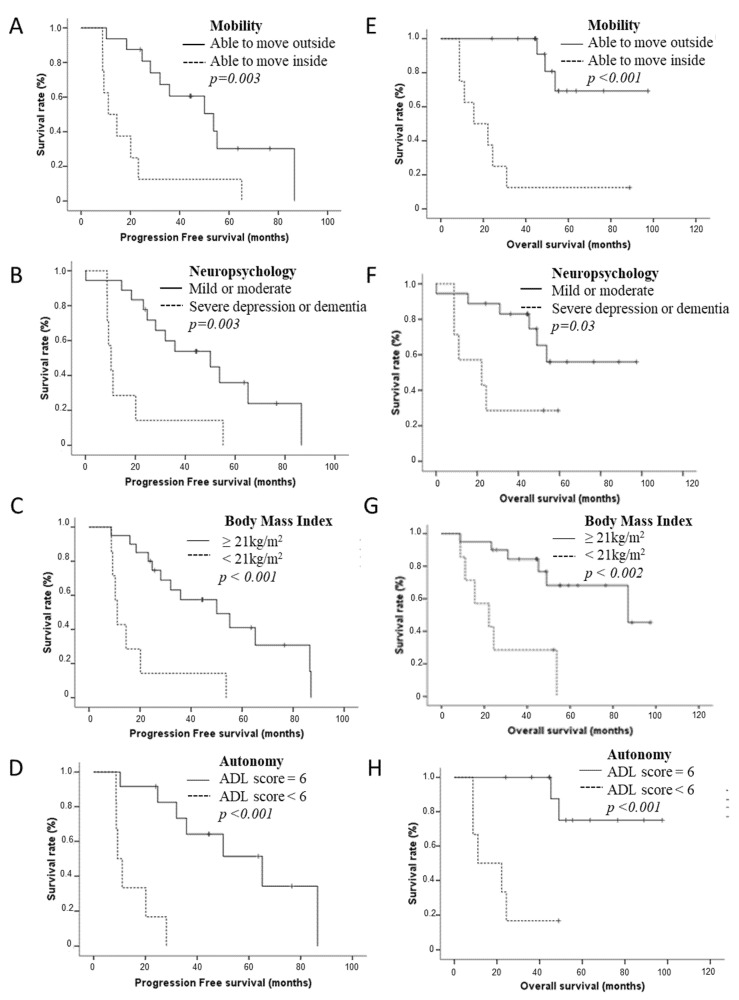
Progression-free survival (**A**–**D**) according to geriatric parameters in elderly patients *IDHm* HGG: motricity (**A**), neuropsychological disorders (**B**), Body Mass Index (BMI) (**C**), and autonomy (**D**). Overall survival (**E**–**H**) according to geriatric parameters in elderly patients *IDHm* HGG: mobility (**E**), neuropsychological disorders (**F**), BMI (**G**), and autonomy (**H**).

**Figure 3 cancers-14-05509-f003:**
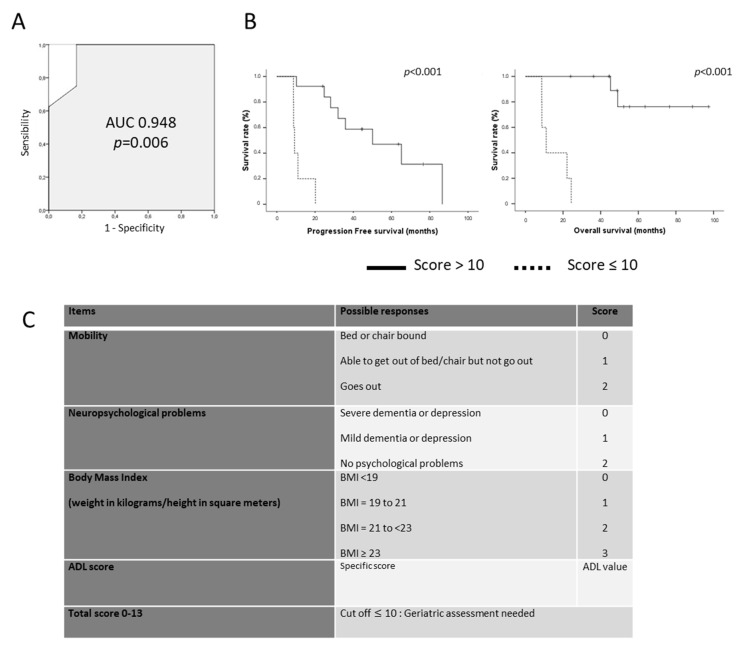
(**A**). ROC (Receiver Operating Characteristics) curve of the Brain Geriatric Score according to long-term survivors. (**B**). Progression-Free Survival (left) and Overall Survival (right) of patients according to the BGS using a cut-off of 10. (**C**). Brain Tumor Geriatric Score detail.

**Table 1 cancers-14-05509-t001:** Baseline characteristics and treatment patterns in elderly patients *IDHm,* elderly patients *IDHwt* and younger patients *IDHm* high-grade glioma.

	Elderly IDHm (*n* = 39)	Elderly IDHwt (*n* = 80)	Younger IDHm (*n* = 919)	IDHm vs. IDHwt Elderly	IDHm Elderly vs. Younger
Characteristics	N (% or Range)	N (% or Range)	N (% or Range)	*p*-Value	*p*-Value
Age (years, median, range)	74.0 (70.2–87.1)	74.4 (70–84.2)	44 (17.1–70)	0.914	0.001
Gender					
Male	22 (56.4)	45 (56.2)	526 (57.2)	0.728	0.830
Female	17 (43.6)	35 (43.8)	393 (42.8)		
Symptoms at diagnosis					
Epilepsy	17 (43.6)	31 (40.3)	546 (61.1)	0.438	0.063
Cognitive disorders	9 (23.1)	25 (32.9)	104 (11.7)	0.045	0.053
Neuro-imaging characteristics					
Contrast enhancement	26 (78.8)	69 (95.8)	551 (69.1)	0.001	0.406
Necrosis	7 (31.8)	40 (78.4)	117 (25.3)	0.022	0.633
WHO 2021 diagnoses					
Anaplastic oligodendroglioma	28 (71.8)	0	496 (54)		
IDHm Anaplastic Astrocytoma	5 (12.8)	2 (2.5)	210 (22.9)		
IDHm grade IV Astrocytoma	6 (15.4)	78 (97.5)	213 (23.1)		
Histological characteristics					
Necrosis	7 (17.9)	41 (51.3)	185 (20.2)	0	0.672
Microvascular Proliferation	28 (71.8)	67 (83.8)	595 (64.9)	0.031	0.554
p53 expression	22 (57.9)	52 (65)	566 (61.9)	0.34	0.527
Ki 67 expression (mean, range)	15 (1–40)	20 (3–90)	15 (1–90)	0.005	0.929
Post-operative KPS (median, range)	80 (50–100)	80 (30–100)	90 (10–100)	0.715	0.005
Corticosteroid intake	16 (48.5)	48 (65.8)	448 (54.6)	0.113	0.661
Extent of resection					
Gross total/Subtotal	9 (24.3)	23 (31)	437 (50.3)	0.746	0.002
Partial	5 (13.5)	10 (13.5)	264 (30.4)		
Biopsy	23 (62.2)	41 (55.4)	167 (19.2)		
Adjuvant Treatment				<0.001	<0.001
Wait and scan policy	3 (7.7)	0	37 (4.2)		
Radiotherapy alone	3 (7.7)	11 (14.9)	114 (13)		
Chemotherapy alone (TMZ or PCV)	16 (41)	11 (14.9)	129 (14.7)		
RT-TMZ	10 (25.6)	46 (62.2)	299 (34)		
RT-PCV	6 (15.4)	1 (1.4)	290 (33)		
Palliative care	1 (2.6)	5 (6.8)	6 (0.7)		

**Table 2 cancers-14-05509-t002:** Baseline geriatric parameters in elderly patients *IDH*-mutated high-grade glioma (*N* = 34).

Parameters	*N*	%
Post-operative KPS ≥ 70	16/21	76
Cognitive disorders		
MMSE ≥ 24	10/11	91
Neuropsychological disorders ^a^	7/25	28
Mobility		
Getting out without assistance	16/24	67
Comorbidities		
Charlson’s index ≥ 5	23/32	72
Medications > 3	17/26	65
Nutrition		
Anorexia	6/24	25
Weight loss	12/25	48
BMI < 21	7/27	26
Autonomy		
ADL < 6	6/18	33.3
IADL < 4	8/17	47
Not a good self-reported state of health	4/17	23.5
G8 score estimation ≤ 14/17	16/25	64

KPS: Karnofsky Performance Scale; MMSE: Mini-Mental State Examination; BMI: Body Mass Index; ADL: Activities of Daily Living; IADL: Instrumental Activities of Daily Living. ^a^ Neuropsychological disorder assessed by G8 score.

**Table 3 cancers-14-05509-t003:** Univariate and multivariate analyses of prognostic factors in elderly patients *IDH* mutated high-grade glioma (*N* = 34). PFS: Progression-free survival; OS: Overall survival; HR: Hazard ratio; BMI: Body Mass Index; ADL: Activities of Daily Living; IADL: Instrumental Activities of Daily Living; KPS: Karnofsky Performance Scale.

Factors	PFS			OS		
	Univariate	Multivariate ^b^		Univariate	Mutivariate ^a^	
	*p* Value	*p* Value	HR	*p* Value	*p* Value	HR
G8 score (items)						
Anorexia (yes vs. no)	0.349			0.511		
Weight loss (yes vs. no)	0.775			0.052		
Mobility (able to get out of chair/bed vs. go out)	0.003	0.018	4.74 (1.30–17.21)	<0.001	0.008	9.93 (1.82–53.98)
Neuropsychological disorders (severe vs. mild or moderate depression or dementia)	0.003	0.005	4.31 (1.56–11.92)	0.036	0.047	3.51 (1.02–12.14)
BMI (≤21 kg/m^2^ vs. > 21 kg/m^2^)	<0.001	0.002	5.26 (1.83–15.04)	0.002	0.006	5.85 (1.66–20.61)
Number of medications > 3 (yes vs. no)	0.186			0.346		
Self-reported state of health (lower vs. similar/better)	0.775			0.047		
Age ranges (>80 vs. <80 years)	0.379			0.741		
Autonomy						
ADL (<6 vs. 6)	<0.001	0.002	14.97 (2.73–81.95)	<0.001	0.010	21.56 (2.11–219.86)
IADL (<4 vs. 4)	0.002			0.132		
Charlson’s index score (≥5 vs. <5)	0.603			0.366		
Biological factors						
Haemoglobin serum level (≤ vs. >median)	0.034	0.200		0.506		
Lymphocyte count ^b^ (≤ vs. >median)	0.009	0.020	4.80 (1.28–18.08)	0.143	0.309	
Neutrophil count (≤ vs. >median)	0.527			0.061		
Platelet count (≤ vs. >median)	0.458			0.863		
Elevation of serum creatinine (≥ vs. <median)	0.588			0.599		
Serum albumin level (≤ vs. >median)	0.641			0.346		
Post-operative KPS (<70 vs. ≥70)	0.012	0.023	3.77 (1.20–11.80)	0.002	0.007	5.30 (1.57–17.95)
Grade (III vs. IV)	0.004			0.001		
Codeletion (yes vs. no)	0.109			0.043		
Type of surgery (biopsy vs. total/subtotal resection)	0.827			0.675		
Steroids intake (<10 vs. ≥10 mg/per day)	0.019			0.380		

^a^ codeletion 1p19q adjusted; ^b^ codeletion 1p19q and steroids taking-adjusted.

## Data Availability

The medical data are not publicly available due to ethical reasons.

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
