# Peer review of "Characteristics, Patterns of Care and Predictive Geriatric Factors in Elderly Patients Treated for High-Grade IDH-Mutant Gliomas: A French POLA Network Study"

_cancers, 2022, doi:10.3390/cancers14225509_

Round 1

Reviewer 1 Report

The present manuscript extensively deals with characteristics and patterns of care of elderly patients affected by HGGs.

I found particularly interesting the data reading the patterns of care, since there are still many unmet needs regarding the best clinical practice in oncological elderly patients. Furthermore,  the Authors highlighted the importance of considering gross total surgery and RT on the basis of global performance status rather than age alone and this is a very modern approach.

I would have been delighted to read more about tumor related epilepsy in this particular cohort of patients. 

Author Response

We thank the reviewers for their constructive comments. We have outlined below (in blue text) our responses to the reviewer comments and revisions to the manuscript. In the revised manuscript, all revisions are highlighted.

Reviewer #1: The present manuscript extensively deals with characteristics and patterns of care of elderly patients affected by HGGs. I found particularly interesting the data reading the patterns of care, since there are still many unmet needs regarding the best clinical practice in oncological elderly patients. Furthermore, the Authors highlighted the importance of considering gross total surgery and RT on the basis of global performance status rather than age alone and this is a very modern approach.

I would have been delighted to read more about tumor related epilepsy in this particular cohort of patients.

We thank the reviewer for his/her comments and question. We analyzed more precisely the tumor related epilepsy in our cohort and added the details in the result section (lines 207-209).

We do hope that this revised version of the manuscript will meet the high standards of the reviewing committee, and that a favorable decision will be made regarding the publication of this article.
Please accept the assurance of our highest consideration.

Reviewer 2 Report

Very elementary paper that show obvious result just in an very uncommon setting like elderly patients with IDH mutated High grade gliomas.

Author Response

We thank the reviewers for their constructive comments. We have outlined below (in blue text) our responses to the reviewer comments and revisions to the manuscript. In the revised manuscript, all revisions are highlighted.

Reviewer #2: Very elementary paper that show obvious result just in an very uncommon setting like elderly patients with IDH mutated High grade gliomas.

We thank the reviewer for his/her comments and effort.

We do hope that this revised version of the manuscript will meet the high standards of the reviewing committee, and that a favorable decision will be made regarding the publication of this article.
Please accept the assurance of our highest consideration.

Reviewer 3 Report

In this manuscript, Montegut and coauthors evaluate a prospectively collected cohort of patients with high grade glioma containing an “oligodendroglial” component on histopathology. 1433 cases were reviewed and included in the POLA cohort, with clinical, pathological, and survival information. This particular manuscript compares the subset of elderly patients (>70yrs) with IDH mutation against elderly patients without IDH mutation, or against young adults with IDH mutation. The strengths of this study are the prospective collection of patient data, the size of the cohort, and the clear writing of the results section and figures. The work is a clear addition to the literature and addresses an important gap for a small subset of patients. There are a few points that could help the authors strengthen their findings:

 One critical finding the authors address in the discussion is the disparity in care received by elderly vs young IDHm patients (lack of XRT in elderly). If numbers are sufficient, it would be interesting to compare survival in elderly vs young IDHm patients who received XRT to see whether the outcome disparity persists. It might also be worth highlighting in the abstract (but will defer to the authors judgement).

 I would also recommend adding the outcome information (possibly another KM plot) for elderly patients without or with IDHmut who did receive radiation. Again, this seems to be a significant point that may affect practice.

 Along the same vein, the authors show elderly IDHm pts receiving RT-PCV lived longer than those receiving RT-TMZ. This is a bit bold given the small sample size. If the authors choose to leave it in, at a minimum they should compare baseline prognostic characteristics between groups (degree of resection, mobility, G8 score, loss of autonomy, denutrition, etc). This will allow readers to decide if they are comparing survival between similar cohorts.

 Two minor points:

-          the numbers for PFS and OS in the abstract differ slightly from the results section. Please double-check and ensure final results are consistent.

-          The introduction could use a little refining and improvement for written English perspective.

Author Response

We thank the reviewers for their constructive comments. We have outlined below (in blue text) our responses to the reviewer comments and revisions to the manuscript. In the revised manuscript, all revisions are highlighted.

In this manuscript, Montegut and coauthors evaluate a prospectively collected cohort of patients with high grade glioma containing an “oligodendroglial” component on histopathology. 1433 cases were reviewed and included in the POLA cohort, with clinical, pathological, and survival information. This particular manuscript compares the subset of elderly patients (>70yrs) with IDH mutation against elderly patients without IDH mutation, or against young adults with IDH mutation. The strengths of this study are the prospective collection of patient data, the size of the cohort, and the clear writing of the results section and figures. The work is a clear addition to the literature and addresses an important gap for a small subset of patients. There are a few points that could help the authors strengthen their findings:

One critical finding the authors address in the discussion is the disparity in care received by elderly vs young IDHm patients (lack of XRT in elderly). If numbers are sufficient, it would be interesting to compare survival in elderly vs young IDHm patients who received XRT to see whether the outcome disparity persists. It might also be worth highlighting in the abstract (but will defer to the authors judgement). We thank the reviewer for this interesting suggestion. We analyzed the survival curves of young and elderly patients receiving radiotherapy but still observed a distinct prognosis, probably related to other treatment disparities like extend of surgery etc… Unfortunately, the number of patients did not allow us to perform a complete multivariate analysis. We added these results in the result section (lines 287-289) and highlight the treatment disparity in the abstract (lines 61-66).

I would also recommend adding the outcome information (possibly another KM plot) for elderly patients without or with IDHmut who did receive radiation. Again, this seems to be a significant point that may affect practice.

We thank the reviewer for this pertinent recommendation. We confirm that IDH mutation was associated with significant survival improvement in elderly patients who received radiotherapy. We added this result in the result section and in the supplementary figures (lines 287-289, Figure S2).

Along the same vein, the authors show elderly IDHm pts receiving RT-PCV lived longer than those receiving RT-TMZ. This is a bit bold given the small sample size. If the authors choose to leave it in, at a minimum they should compare baseline prognostic characteristics between groups (degree of resection, mobility, G8 score, loss of autonomy, denutrition, etc). This will allow readers to decide if they are comparing survival between similar cohorts.

We agree with the reviewer regarding the limitations of this analysis that we planned to add at supplementary figure and data. only Following his/her recommendations we added the patient characteristic table of all patients included in this sub-analysis (Tables S5).

Two minor points:

the numbers for PFS and OS in the abstract differ slightly from the results section. Please double-check and ensure final results are consistent.

We thank the reviewer for this notification. We corrected it.

The introduction could use a little refining and improvement for written English perspective.

We agree with the reviewer and send the manuscript for English correction before resubmission.

We do hope that this revised version of the manuscript will meet the high standards of the reviewing committee, and that a favorable decision will be made regarding the publication of this article.
Please accept the assurance of our highest consideration.

Round 2

Reviewer 2 Report

This paper have obvious result, is overdiscussed on the basis of results obtained based on very few and really rare subets of Ederly patients with high grade primary CNS tumors.